# Prediction of Thermo-Physical Properties of TiO_2_-Al_2_O_3_/Water Nanoparticles by Using Artificial Neural Network

**DOI:** 10.3390/nano10040697

**Published:** 2020-04-07

**Authors:** Milad Sadeghzadeh, Heydar Maddah, Mohammad Hossein Ahmadi, Amirhosein Khadang, Mahyar Ghazvini, Amirhosein Mosavi, Narjes Nabipour

**Affiliations:** 1Department of Renewable Energy and Environmental Engineering, University of Tehran, Tehran 1439957131, Iran; milad.sadeghzadeh@gmail.com; 2Department of Chemistry, Payame Noor University (PNU), Tehran P.O. Box, 19395-3697, Iran; heydar.maddah@gmail.com (H.M.); Amirhossein.khadang77@gmail.com (A.K.); 3Faculty of Mechanical Engineering, Shahrood University of Technology, POB- Shahrood 3619995161, Iran; 4Department of Ocean and Mechanical Engineering, Florida Atlantic University, 777 Glades Road Boca Raton, FL 33431, USA; m.ghazvini@alumni.ut.ac.ir; 5Kalman Kando Faculty of Electrical Engineering, Obuda University, 1034 Budapest, Hungary; 6Institute of Structural Mechanics (ISM), Bauhaus-Universität Weimar, 99423 Weimar, Germany; 7Thuringian Institute of Sustainability and Climate Protection, 07743 Jena, Germany; 8Department of Mathematics and Informatics, J. Selye University, 94501 Komarno, Slovakia; 9Institute of Research and Development, Duy Tan University, Da Nang 550000, Viet Nam

**Keywords:** thermal conductivity, TiO_2_-Al_2_O_3_/water, nanofluid, artificial neural network

## Abstract

In this paper, an artificial neural network is implemented for the sake of predicting the thermal conductivity ratio of TiO_2_-Al_2_O_3_/water nanofluid. TiO_2_-Al_2_O_3_/water in the role of an innovative type of nanofluid was synthesized by the sol–gel method. The results indicated that 1.5 vol.% of nanofluids enhanced the thermal conductivity by up to 25%. It was shown that the heat transfer coefficient was linearly augmented with increasing nanoparticle concentration, but its variation with temperature was nonlinear. It should be noted that the increase in concentration may cause the particles to agglomerate, and then the thermal conductivity is reduced. The increase in temperature also increases the thermal conductivity, due to an increase in the Brownian motion and collision of particles. In this research, for the sake of predicting the thermal conductivity of TiO_2_-Al_2_O_3_/water nanofluid based on volumetric concentration and temperature functions, an artificial neural network is implemented. In this way, for predicting thermal conductivity, SOM (self-organizing map) and BP-LM (Back Propagation-Levenberq-Marquardt) algorithms were used. Based on the results obtained, these algorithms can be considered as an exceptional tool for predicting thermal conductivity. Additionally, the correlation coefficient values were equal to 0.938 and 0.98 when implementing the SOM and BP-LM algorithms, respectively, which is highly acceptable.

## 1. Introduction

Recently, numerous endeavors have been made in order to enhance the performance of various applications with the help of nanotechnology [1,2,3,4]. As an illustration, it is practicable to reduce system size or enhance the thermal performance of materials [5,6,7,8]. In this way, some investigations have been implemented on the use of nanotechnology in thermal applications [9,10,11,12,13,14,15,16,17,18,19]. Additionally, some studies have focused on the prediction of the thermal conductivity ratio associated with various nanofluids with the help of using experiments and artificial neural networks [20,21,22,23,24,25,26,27,28,29,30,31]. Vafaei et al. [32] predicted the thermal conductivity ratio of MgO-MWCNTs/EG hybrid nanofluids by using ANN (artificial neural network) at the temperature range of 25–50 °C. According to the results, the best performance belonged to the neural network with 12 neurons in the hidden layer. Also, an investigation has been carried out by Afrand et al. [33] to estimate the thermal conductivity of MgO/water nanofluid. Furthermore, by implementing an ANN, convective heat transfer of TiO_2_/water nanofluid has been studied by Esfe et al. [34]. As indicated in the results, the regression coefficient of the model for the Nusselt number’s data is 99.94%. Azizi et al. [35] employed ANN to estimate the water holdup in different layouts of oil-water two-phase flow. In another use of ANN, Azizi et al. [36] investigated the estimation of void fraction in pipes with different inclination. ANN-based methods have this potential to give high precision estimation which can be beneficial in real practice since the actual experiment is not only so expensive but also very time-consuming.

On the other hand, the sol–gel process involving hydrolysis and condensation reactions of alkali precursors is an appropriate method for the synthesis of ultra-fine metal oxide [37]. Different researchers have used the sol–gel method in different conditions. Li et al. [38] added tetra-n-butyl titanate to deionized water and hydrochloric acid or ammonia. After milling and drying the gel at different temperatures, TiO_2_ nanopowder was obtained. Zhang et al. [39] used the sol–gel microemulsion method. They synthesized TiO_2_ nanoparticles by hydrolysis of tetraizo titanium Prop Oxide with 80 Tween-Span in a microemulsion and then calcined it is at different temperatures. The results show that the particles are spherical. In some cases, the surfactant is used in the sol–gel process. Pavasupree et al. [40] synthesized semi-porous TiO_2_ nanoparticles by adding hydrochloride clarinylamine (LAHC) as a surfactant to the precursor solution. The resulting powders were calcined for 4 h at 400 °C. In the same way, Colon et al. [41] increased the specific surface area of the particles by adding activated carbon to the solution. The XRD results showed only the presence of the anatase phase in the powders. Li et al. [37] aged the gel for 12 h at 100 °C after drying it. The results showed that aging help to remove organic compounds and influence atomic penetration and crystalline anatase. In 2014, SiO_2_ nanoparticles were synthesized by Oliveira et al. They used the polypropylene matrix in their research. Their results showed that the production of inorganic nanoparticles in a polymer solution does not require solvent through the reaction in the molten phase [42]. Moreover, the influence of adding Al_2_O_3_ and TiO_2_ nanoparticles into the drilling mud was studied by Ghasemi et al. [43]. The size of Al_2_O_3_ and TiO_2_ nanoparticles were 20 and 60 nm, respectively, and a concentration of 0.05 wt. %. Based on the obtained results of temperature and pressure effects, the drilling mud rheological properties such as plastic viscosity are decreased by increasing the temperature, nonetheless, the pressure rise augments these properties. Additionally, the influences of pressure in low temperature outweighs in high temperatures. Also, the effective electrical conductivity of Al_2_O_3_ nanoparticles was experimentally measured by Ganguly et al. [44]. For examining the influences of temperature variations and volume fraction on the electrical conductivity of Al_2_O_3_ nanofluids, experiments have been carried out as a function of these parameters. As indicated in the results, the electrical conductivity increases significantly with augmenting volume fraction and temperature. Nonetheless, the effective conductivity’s reliance on the volume fraction is much higher than the temperature. Furthermore, some investigations have been intensively carried out for increasing the nanofluids’ thermal conductivity with the help of applying different kinds of nanoparticles [45,46]. 

The aim of this study is to investigate the thermo-physical properties of TiO_2_-Al_2_O_3_ nanoparticles in water that can be employed as a coolant fluid with its improved thermal properties. This is accomplished by conducting experiments on various volumes of nanoparticles in water. In this study, special attention has been paid to the temperature effect on the nanofluid’s thermal conductivity. The temperature’s influence on the thermal conductivity of TiO_2_ nanofluid has not been reported yet. Furthermore, the current investigation discloses the influence of temperature and nanoparticle concentrations on the thermal properties of hybrid nanofluids. With the help of the experimental results obtained by this study, researchers can acquire exceptional information regarding the displacement of nanofluid and its properties, in which appropriate theoretical models can be achieved in the future.

## 2. Test section

### Synthesis of TiO_2_-Al_2_O_3_ Nanoparticles and Characterization

In this study, the sol–gel method was used to synthesize TiO_2_-Al_2_O_3_ nanoparticles prepared in various percentages of Al_2_O_3_ (10–60). Two different solution samples were prepared for this nanofluid. In the first sample, 0.1105 mol (2 g) of TiCl_4_ was dissolved in a solution that contains 10 mL of methyl acetate, 10 mL of ethanolamine and 100 mL of ethanol, and stirred for one hour at room temperature. Finally, a uniform suspension was produced. Then, AlCl_3_ was added to the solution in various weights (0–100%) and the resulting solution was stirred for one hour at 80 °C. The second sample solution was made up of 30 mL of n-hexane, 20 mL of ethanol, 4 mL of methyl acetate and 5 mL of ammonium hydroxide. The second sample was added to the first sample and the solution was mixed simultaneously to obtain the hydrogel. By adding the second sample, the viscosity of the hydrogel increased. After the addition of the sample was complete, the solution was stirred at room temperature for 48 h and then kept at room temperature for 12 h. After 12 h, with the help of water, the obtained gel was washed to remove the chloride salts and then separated solids from it. The solids were washed three times with distilled water and then placed in an oven for 3 h at 900 °C. Figure 1 shows the schematic of nanocomposite synthesis.

The preparation of nanofluid is the first step in changing the heat transfer efficiency. The preparation of nanofluid by adding the nanoparticles to the base fluid should not be considered as a solid–liquid mixture. Because the preparation of nanofluid requires special conditions. Some of these special conditions include uniform and stable suspension, aggregation of particles and the lack of change in the nature of the base fluid. Different methods are used to achieve these specific properties. Various concentrations of nanofluids are prepared by using the equation below:(1)$$\varphi \;\left(Volume\;\mathrm{concentration}\%\right)=\frac{\frac{w_{np}}{\rho_{np}}}{\frac{w_{np}}{\rho_{np}}+\frac{w_{water}}{\rho_{water}}}$$
where $\rho_{np}$ and $\rho_{water}$ represent the nanoparticles’ density and water density, respectively. w is their mass [5].

Nanofluids that are prepared by the two-step method should be stable and the particle should not be sedimented in the fluid. Therefore, the nanofluid’s stabilization should be considered.

In this study, a magnetic stirrer was used for nanofluid stability. The agitation intensity is crucially important for the nanoparticles’ dispersion. The particles are connected to each other through bonds and the weak bonds are broken with force. However, there is a forceful propensity in nanoparticles for agglomerating because of the van der Waals force. We used the TEM analysis to evaluate the produced nanoparticles (Electro Microscopy) (PHILIPS EM 208, FEI, Hillsboro, Oregon, USA). SEM analysis was used to evaluate the morphology of synthesized nanoparticles. Figure 2 shows the SEM of pure nanoparticles and nanocomposites.

According to the above, 20% alumina and 80% titanium were selected as samples for heat transfer analysis (the main objective of this study is to improve the properties of TiO_2_). Therefore, adding more alumina will keep TiO_2_ away from its main. With the help of the TCi Thermal Conductivity analyzer made by Canada’s C-Therm, the thermal conductivity of the nanocomposite has been calculated experimentally. Also, the Brookfield Viscometer was used to measure the viscosity of prepared nanofluid. Based on the manufacturer and the obtained results, the proposed approach for measuring thermal conductivity brings an uncertainity of ±2% with the deviation of 4% for each measurement. The repeatability and accuracy of the viscometer used are ±0.2% and ±1% in the full-scale range (FSR) of measurements, respectively. One noteworthy approach in the field of thermal analysis is differential scanning calorimetry (DSC). This approach can be found in ASTM E1269. The ASTM E1269 is the standard defined procedure for measuring specific heat capacity through DSC approach. In this research, the improved modulated-DSC approach is used to obtain the specific heats. In modulated-DSC, a sinusoidal temperature fluctuation is employed instead of a linear ramp. This novel technique is capable to calculate the heat capacity and the heat flow of the samples, simultaneously.

## 3. Results and Discussion

As shown in Figure 3, the specific heat capacity of the nanocomposite varied linearly within the range 300–360 K. For temperatures of 300 K, the nanocomposite has a higher thermal capacity than its components. At this temperature, the nanocomposite’s average heat capacity was 0.75 J/gK. In the range of 300–360 K, the average heat capacity was 0.78 J/gK. Since the temperature of the heat transfer analysis was mainly in this range, this number was chosen as the basis for our calculation.

Based on the results, it can be said that in the initial intervals, the thermal conductivity of the nanocomposite was within the range of its components. Because of the large number of Al_2_O_3_ particles in the nanocomposite, the thermal conductivity of the nanocomposite was very close to that of Al_2_O_3_. Based on Figure 4, the thermal conductivity coefficient was calculated to be 11.7 W/mK within the range of 300–360 K.

The variations in the thermal conductivity of the nanofluid with respect to concentration are shown in Figure 5. According to Figure 5, with increasing nanofluid concentration, thermal conductivity also increased. It should be noted that overconcentration may be due to the agglomeration of particles and the reduction of the thermal conductivity of the nanofluid. Increasing the temperature leads to enhanced thermal conductivity. This is due to Brownian motion and an increase in the collision of the particles with each other. Since the presented equations are not based on nanocomposite in nanofluid, or the base fluid is not combined, it is not possible to match the data with this equation. Here, the experimentally measured data is fit on the basis of temperature and concentrations to be employed in heat transfer analysis.

The relationships among the temperature, concentration, and thermal conductivity of the nanofluid were obtained on the basis of the experimental data. In Figure 5 and Figure 6, the 3D contour illustrates the predicted data resulting from the experimental data. In the equations, x represents the concentration, y represents the temperature, and z is the thermal conductivity coefficient.

Linear model Poly55: F(x, y) = p00 + p10 × x + p01 × y + p20 × x^2^ + p11 × x × y + p02 × y^2^ + p30 × x^3^ + p21 × x^2^ × y + p12 × x × y^2^ + p03 × y^3^ + p40 × x^4^ + p31 × x^3^ × y + p22 × x^2^*y^2^ + p13 × x × y^3^ + p04 × y^4^ + p50 × x^5^ + p41 × x^4^ × y + p32 × x^3^ × y^2^ + p23 × x^2^ × y^3^ + p14 × x × y^4^ + p05 × y^5^(2)

Coefficients (with 95% confidence bounds):
p10 = 3.494 × 10^−1^ (1.966 × 10^−1^, 5.022 × 10^−1^)p01 = 7.815 × 10^−3^ (−1.715 × 10^−2^, 3.279 × 10^−2^)p20 = −3.843 × 10^−1^ (−8.201 × 10^−1^, 5.145 × 10^−2^)p11 = −1.749 × 10^−2^ (−2.926 × 10^−2^, −5.721 × 10^−3^)p02 = −1.736 × 10^−4^ (−1.623 × 10^−3^, 1.276 × 10^−3^)p30 = 4.467 × 10^−1^ (−2.094 × 10^−1^, 1.103)p21 = 8.337 × 10^−3^ (−1.159 × 10^−3^, 1.783 × 10^−2^)p12 = 4.972 × 10^−4^ (7.729 × 10^−5^, 9.171 × 10^−4^)p03 = 1.163 × 10^−6^ (−3.839 × 10^−5^, 4.071 × 10^−5^)p40 = −3.24 × 10^−1^ (−7.665 × 10^−1^, 1.186 × 10^−1^)p31 = −4.751 × 10^−3^ (−1.065 × 10^−2^, 1.146 × 10^−3^)p22 = −4.931 × 10^−5^ (−2.011 × 10^−4^, 1.025 × 10^−4^)p13 = −7.265 × 10^−6^ (−1.401 × 10^−5^, −5.195 × 10^−7^)p04 = 1.758 × 10^−8^ (−4.95 × 10^−7^, 5.302 × 10^−7^)p50 = 8.486 × 10^−2^ (−2.493 × 10^−2^, 1.946 × 10^−1^)p41 = 2.198 × 10^−3^ (5.41 × 10^−4^, 3.855 × 10^−3^)p32 = −2.4 × 10^−5^ (−6.092 × 10^−5^, 1.293 × 10^−5^)p23 = 5.426 × 10^−7^ (−5.136 × 10^−7^, 1.599 × 10^−6^)p14 = 3.898 × 10^−8^ (−1.76 × 10^−9^, 7.971 × 10^−8^)p05 = −2.131 × 10^−10^ (−2.756 × 10^−9^, 2.33 × 10^−9^)p00 = 4.794 × 10^−1^ (3.199 × 10^−1^, 6.388 × 10^−1^)

Figure 7 shows the effect of the increase of nanoparticles on the viscosity of the base fluid. As shown in Figure 7 and Figure 8, as the nanofluid concentration increases, the viscosity increases. These solid particles in the base fluid increased the collision of particles, leading to an increase in viscosity. The equation and the corresponding graph for viscosity are presented below.

Linear model Poly55:
f(x, y) = p00 + p10 × x + p01 × y + p20 × x^2^ + p11 × x × y + p02 × y^2^ + p30 × x^3^ + p21 × x^2^ × y + p12 × x × y^2^ + p03 × y^3^ + p40 × x^4^ + p31 × x^3^ × y + p22 × x^2^ × y^2^ + p13 × x × y^3^ + p04 × y^4^ + p50 × x^5^ + p41 × x^4^ × y + p32 × x^3^ × y^2^ + p23 × x^2^ × y^3^ + p14 × x × y^4^ + p05 × y^5^(3)

Coefficients (with 95% confidence bounds):
p00 = 1.043 × 10^−3^ (1.442 × 10^−4^, 1.941 × 10^−3^)p10 = −1.066 × 10^−4^ (−9.675 × 10^−4^, 7.544 × 10^−4^)p01 = 4.699 × 10^−5^ (−9.37 × 10^−5^, 1.877 × 10^−4^)p20 = −5.178 × 10^−4^ (−2.973 × 10^−3^, 1.937 × 10^−3^)p11 = 4.561 × 10^−5^ (−2.071 × 10^−5^, 1.119 × 10^−4^)p02 = −3.333 × 10^−6^ (−1.15 × 10^−5^, 4.832 × 10^−6^)p30 = 4.982 × 10^−4^ (−3.198 × 10^−3^, 4.195 × 10^−3^)p21 = 2.274 × 10^−5^ (−3.076 × 10^−5^, 7.625 × 10^−5^)p12 = −2.283 × 10^−6^ (−4.649 × 10^−6^, 8.235 × 10^−8^)p03 = 6.58 × 10^−8^ (−1.57 × 10^−7^, 2.886 × 10^−7^)p40 = −2.457 × 10^−4^ (−2.739 × 10^−3^, 2.248 × 10^−3^)p31 = −1.03 × 10^−5^ (−4.352 × 10^−5^, 2.292 × 10^−5^)p22 = −3.773 × 10^−7^ (−1.233 × 10^−6^, 4.781 × 10^−7^)p13 = 4.347 × 10^−8^ (5.466 × 10^−9^, 8.147 × 10^−8^)p04 = −4.355 × 10^−10^ (−3.324 × 10^−9^, 2.453 × 10^−9^)p50 = 4.998 × 10^−5^ (−5.686 × 10^−4^, 6.685 × 10^−4^)p41 = 1.876 × 10^−6^ (−7.46 × 10^−6^, 1.121 × 10^−5^)p32 = 6.451 × 10^−8^ (−1.435 × 10^−7^, 2.725 × 10^−7^)p23 = 2.166 × 10^−9^ (−3.785 × 10^−9^, 8.116 × 10^−9^)p14 = −2.81 × 10^−10^ (−5.105 × 10^−10^, −5.151 × 10^−11^)p05 = 1.276 × 10^−13^ (−1.42 × 10^−11^, 1.445 × 10^−11^)Goodness of fit:SSE: 2.306 × 10^−9^R-square: 0.9991Adjusted R-square: 0.9979RMSE: 1.24 × 10^−5^

The data were obtained from the experiment that were necessary to find a favorable relation between output and input data, and these were the volumetric concentration and temperature of the fluid, respectively. Table 1 presents the range of each of these input parameters.

The nanofluid’s thermal conductivity with respect to the base fluid is considered to be its thermal conductivity ratio, which is an appropriate measurement of nanoparticles with respect to thermal conductivity. As an illustration in SOM, competitive learning methods were developed based on particular properties of the human brain and were used for training. The arrangement of the human brain’s cells in a distinct area is precise and meaningful. As an example, the sensory inputs of hearing or touch can contribute to an important geometric arrangement in distinct regions. Furthermore, processor units are located in nodes in SOM. By considering input patterns, units are arranged in a competitive learning approach. The units’ position is arranged so that a useful coordinate system is created. Thus, a topographic map of the input patterns is created by the SOM in which units’ position is associated with the input patterns’ intrinsic characteristics. As illustrated in Figure 9, the base fluid’s hexagonal arrangement leads to the prediction of the nanofluid’s thermal conductivity ratio. The arrangement of the number of neurons is implemented in 9 × 9 shapes. The overall number associated with the neurons used is 81, with a neuron winner of 9 data. Thus, here, the neuron number 76 is the winner.

Radial basic networks are another kind of neural network. The comparison of baseline radius and post back networks indicates that the former requires a greater number of neurons, and the design of the latter requires more time. The performance of the former is exceptional under conditions in which there are very large educational vectors. Meanwhile, the input layer does not perform any processing. On the other hand, the hidden or second layer performs a significant part in converting nonlinear patterns to linear separation patterns. Finally, in order to find an approximation, a summation function with a linear output is produced by the third layer. Based on Figure 10, the correlation coefficient value was equal to 0.93875, which is auspicious. 

A BP-LM network training algorithm with a two-layered neural network was used for modeling. In this way, the nanoparticle size, volumetric concentration, and temperature were chosen as input data. Furthermore, the thermal conductivity ratio coefficient was employed as the target parameter. The sensitivity analysis is required for the number of neurons in the hidden layer. To achieve this purpose through application of the trial and error method, the quantity of neurons associated with the hidden layer was studied. As indicated in the Figure 10, the best performance belonged to the network with 76 neurons in the hidden layer. The reason for which numbers of neurons greater than 76 are not more attractive is that augmentation of the number of neurons increases the runtime as well as intensifying the possibility error in the model, even though exceptional outcomes are sometimes achieved by augmenting the number of neurons. According to Figure 11, the quantity of 0.98 expresses the achieved correlation coefficient of the thermal conductivity ratio. The overall obtained data are placed within the circumference of the diameter line. The correlation coefficient can be enumerated as the most crucial predictive factor, such that better predictions can be made when this value is closer to 1. It can be clearly observed that the predicted and experimental data can be easily fitted, which is evidence of exceptionally favorable network prediction using 76 neurons.

## 4. Conclusions

Firstly, the TiO_2_-Al_2_O_3_ nanocomposite was synthesized. For the synthesis of nanocomposites, the sol–gel method and TiCl_4_ and AlCl_3_ compounds were used. The results of the analysis showed that all synthesized samples had dimensions in the nano range. After the synthesis of the nanocomposites, they were characterized by TEM. Adding alumina had a significant effect on the TiO_2_ crystal size. The main reason for this is the formation of a homogeneous mixture of Ti-O-Al bonds during the sol–gel process. DSC (differential scanning calorimetry) was used to measure the specific heat capacity of the nanofluid. The nanocomposite showed a higher thermal capacity than its components at 300 K. A TC-Thermal Conductivity Analyzer (C-Therm Canada) was used to measure the thermal conductivity of the nanofluid-containing nanocomposite. The results showed that the average thermal conductivity was 11.7 W/mK. It should be noted that as the concentration of nanofluid increases, the agglomeration of particles also increases; as a result, the thermal conductivity of the nanofluid decreased. An increase in temperature also increases the thermal conductivity coefficient. Based on the experimental data, the relationships among concentration, temperature, thermal conductivity and viscosity were obtained. Finally, neural networks were used to predict the electrical properties of the nanofluid. For this purpose, a neural network with a multilayer perceptron structure was used to develop a model for estimating the thermal properties of nanofluids. In the end, the neural network was able to predict thermal properties by a correlation coefficient of 98%.

## Figures and Tables

**Figure 1 nanomaterials-10-00697-f001:**
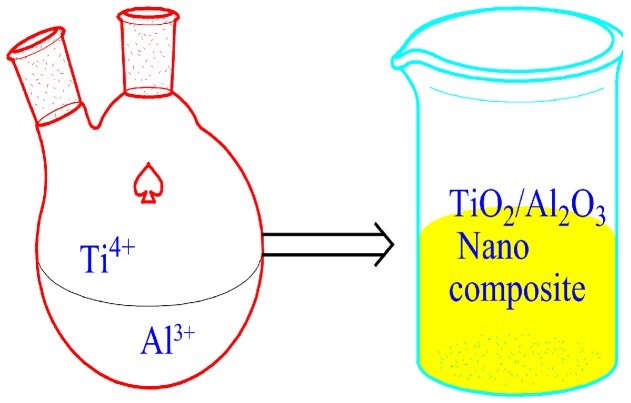
Schematic of nanocomposite synthesis.

**Figure 2 nanomaterials-10-00697-f002:**
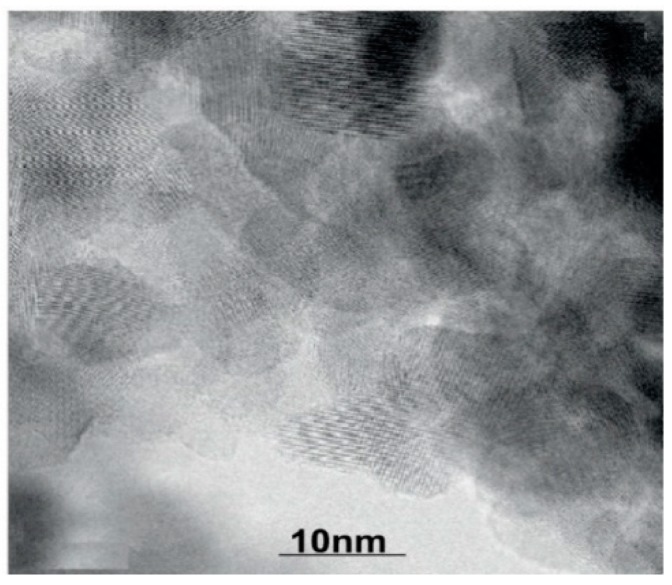
SEM images of nanoparticles after dispersion (20% alumina and 80% titanium).

**Figure 3 nanomaterials-10-00697-f003:**
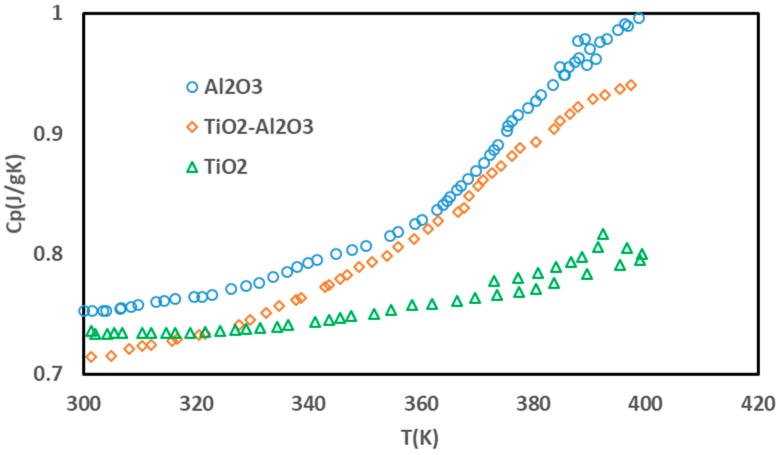
The results of the specific heat capacity of the nanocomposite.

**Figure 4 nanomaterials-10-00697-f004:**
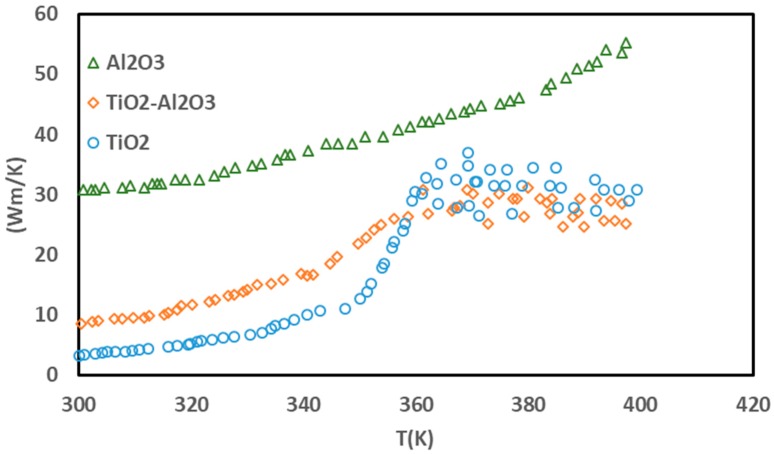
The results of the thermal conductivity coefficient of the nanocomposite.

**Figure 5 nanomaterials-10-00697-f005:**
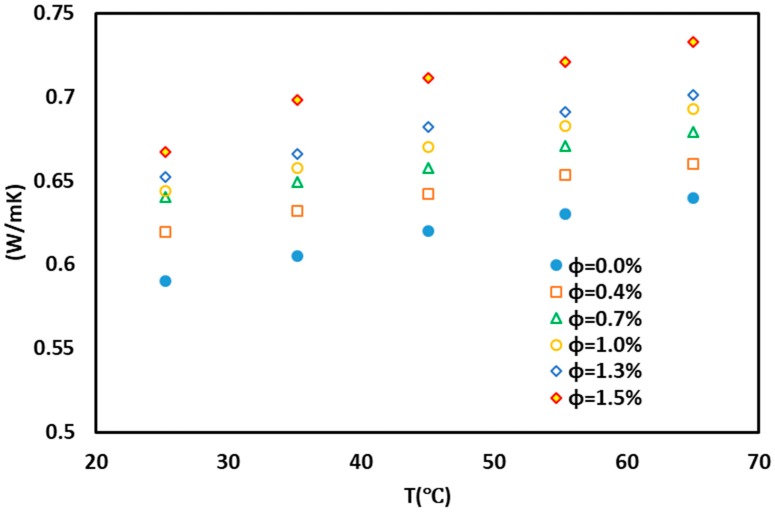
Variations in thermal conductivity coefficient of nanofluid with temperature and concentration of nanoparticles.

**Figure 6 nanomaterials-10-00697-f006:**
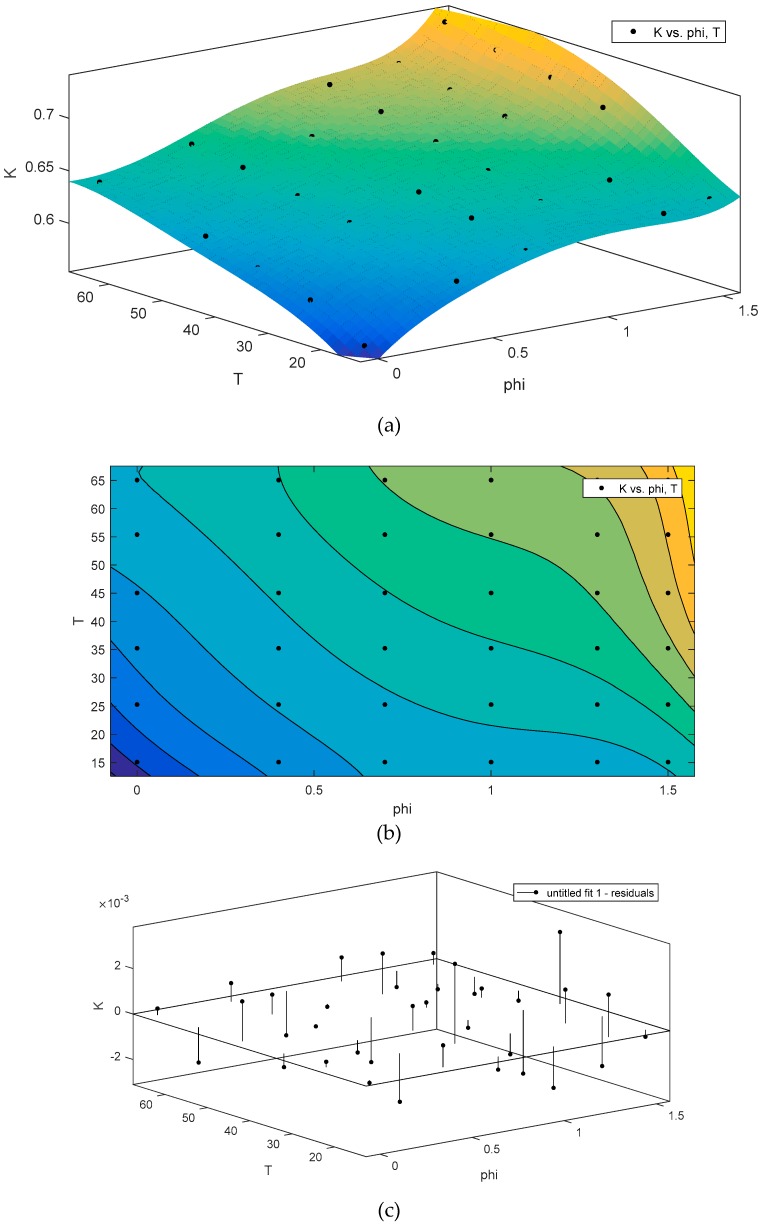
(**a**) Contour plots, (**b**) 3D, and (**c**) proposed model for the thermal conductivity coefficient (Temperature is in °C and phi (Volumetric Concentration (%)).

**Figure 7 nanomaterials-10-00697-f007:**
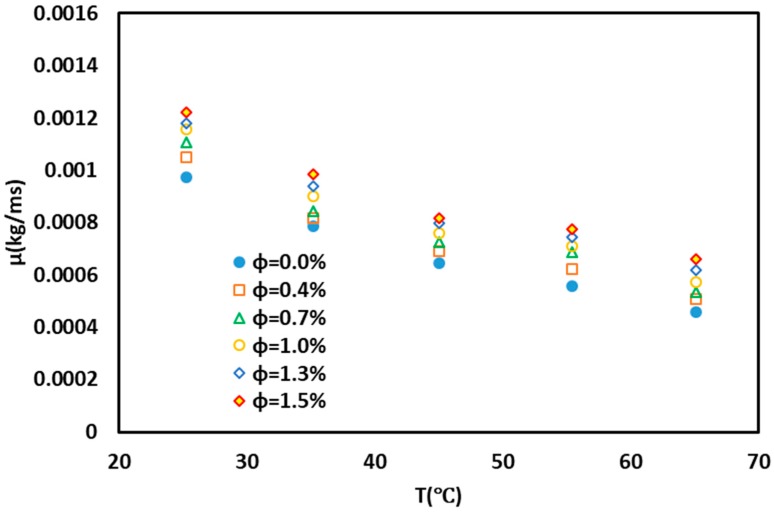
Viscosity variations of nanofluid with temperature and nanoparticle concentrations.

**Figure 8 nanomaterials-10-00697-f008:**
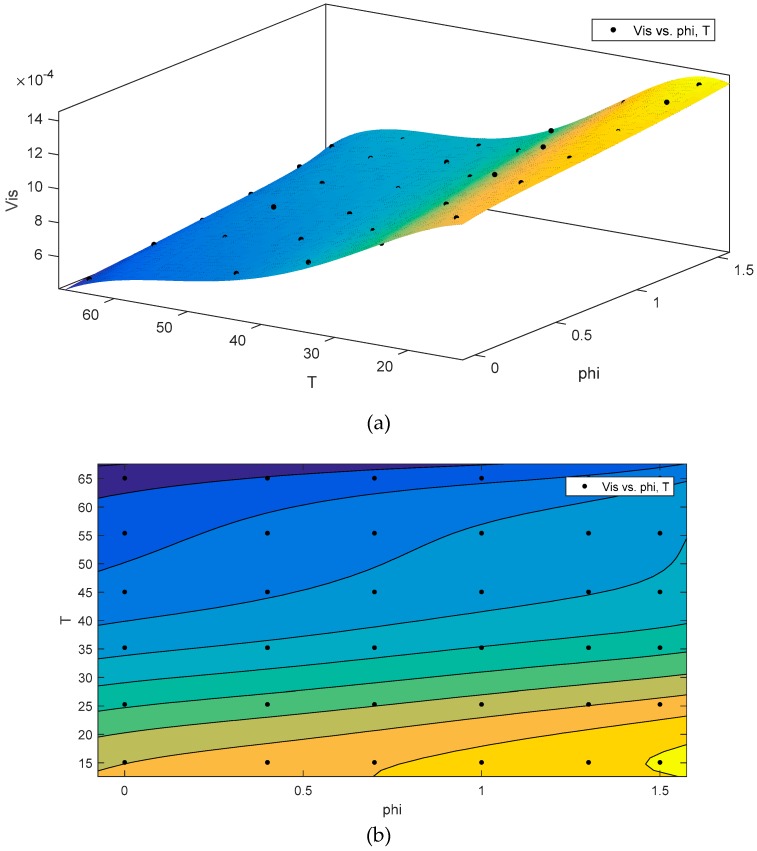
Contour graphs (3D) describing the model’s (**a** to **c**) viscosity (Vis) distribution. (T (temperature) and phi (Volumetric Concentration (%)).

**Figure 9 nanomaterials-10-00697-f009:**
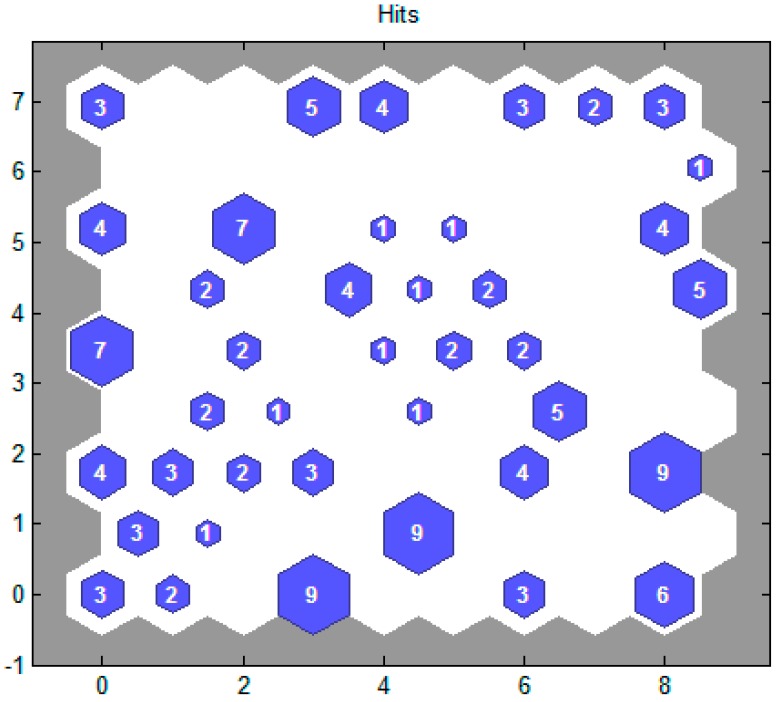
The structure of the neurons used and the quantity of assigned data.

**Figure 10 nanomaterials-10-00697-f010:**
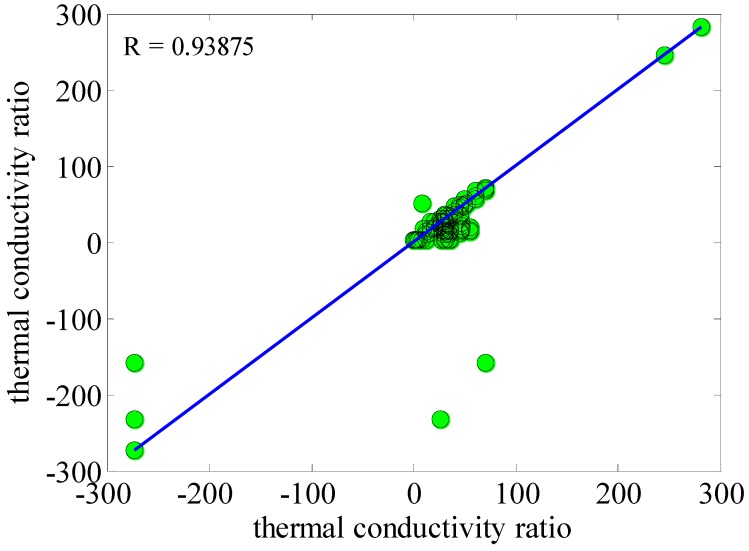
Correlation coefficient data based on investigating the predicted and experimental thermal conductivity ratio.

**Figure 11 nanomaterials-10-00697-f011:**
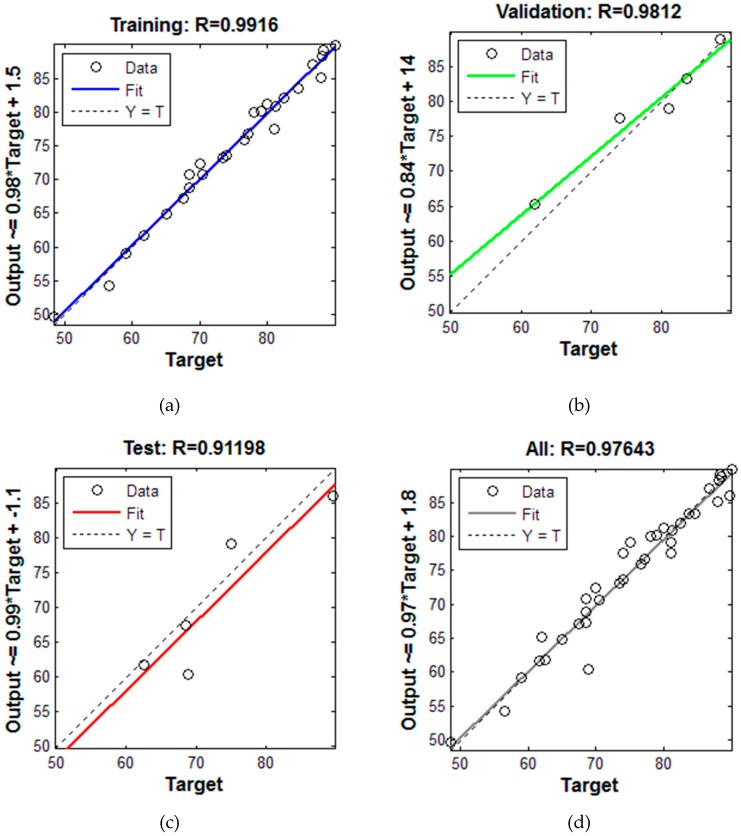
Results based on the correlation coefficient of thermal conductivity ratio. (**a**) Training, (**b**) Validation, (**c**) Test, (**d**) totally).

**Table 1 nanomaterials-10-00697-t001:** Input parameters’ range.

Parameter	Range
Temperature (°C)	10–70
Volumetric Concentration (%)	0.25–6

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
