# Peer review of "Prediction of Thermo-Physical Properties of TiO2-Al2O3/Water Nanoparticles by Using Artificial Neural Network"

_nanomaterials, 2020, doi:10.3390/nano10040697_

Round 1

Reviewer 1 Report

This study is about the thermal properties of TiO2-Al2O3/water nanoparticles using artificial neural network (ANN). Here are some suggestions and comments to improve the manuscript for further process.

How did the authors measure ρnp and ρw? More explanation of DSC is needed in the manuscript. Why is the experimental data of TiO2 and TiO2-Al2O3 turbulent around 360 K in Fig.4 ? What is “wikimeter”? Is that a viscometer? The authors developed the polynomial surface model in degree of 5. Why did the authors choose degree of 5 in the explanatory variable? Does the SOM mean the self-organizing map? More explanation of the relationship between the SOM and the ANN is needed. How did the authors determine the initial condition of the SOM? The reviewer could not understand Fig. 9 and how the authors developed the ANN based on the SOM. What is the structure of the ANN in this model? The authors explained that the two-layered neural network was used with 76 neurons although the SOM was used to predict the polynomial surface model. It is not enough to explain line229-243 in the manuscript.

Author Response

Dear Editor and Reviewers,

We appreciate very much the editor and the reviewers for the constructive comments. We also thank the editor and the reviewers for the effort and time put into the review of the manuscript. Each comment has been carefully considered point by point and responded. Responses to the reviewers and changes in the revised manuscript are as follows.

Title: Prediction of Thermo-physical Properties of TiO2-Al2O3/water Nanoparticles by Using Artificial Neural Network

Reviewer 1:

  1. How did the authors measure ρnp and ρw? More explanation of DSC is needed in the manuscript.

Response: Thanks for your valuable comments. Densities has been measured with density meter. Differential scanning calorimetry (DSC) is an invaluable method of thermal analysis in the material sciences. More details are described in ASTM E1269, the standard test method for determining specific heat capacity by DSC. In this study, the specific heats were measured using modulated differential scanning calorimetry, which is a recently developed extension of DSC. It uses a sinusoidal temperature oscillation instead of the traditional linear ramp, which provides the heat capacity of the sample and the heat flow at the same time.

Why is the experimental data of TiO2 and TiO2-Al2O3 turbulent around 360 K in Fig.4?

Response: Thanks for your comment. The total energy of nanoparticle as a function of temperature is discrete at around 360 point longitudinally in the range of latent heat due to the first-order phase transition. Such longitudinal discontinuity due to the latent heat causes a sharp peak in the heat capacity with respect to temperature.

  1. What is “wikimeter”? Is that a viscometer?

Response: It was an error which is corrected in the revised manuscript: “Brookfield Viscometer”

  1. The authors developed the polynomial surface model in degree of 5. Why did the authors choose degree of 5 in the explanatory variable?

Response: Thanks for your comment. The best model with highest coefficient of determination was achieved only for polynomial surface model in degree of 5.

  1. Does the SOM mean the self-organizing map? More explanation of the relationship between the SOM and the ANN is needed. How did the authors determine the initial condition of the SOM?

Response: Yes, it is right and the SOM stands for self-organizing map. In addition, more explanation of the relationship between the SOM and the ANN is added.

  1. The reviewer could not understand Fig. 9 and how the authors developed the ANN based on the SOM. What is the structure of the ANN in this model? The authors explained that the two-layered neural network was used with 76 neurons although the SOM was used to predict the polynomial surface model. It is not enough to explain line229-243 in the manuscript.

Response: Thanks for your valuable comment. SOM-ANN was applied on theral ondutivity data set, having 76 elements (2 variables, 38 samples). The model was validated using real time data obtained from different samples. The maximum iteration was 200 having learning rate .5 and training function trainbu, based on weight and bias learning rules. Weights were adjusted by back-propagation according to training and learning rule. Six significant clusters were obtained by applying grid size of 2 × 2 on 76 elements grouping the temperature and onentration upon the similarities among the elements. Topology used for neighborhood and distance measure between nodes is hexagonal. Input (variables) is presented to all nodes and response (output or cluster) is calculated. Highest response neuron is winning neuron. Weights are adjusted for winning and neighboring neurons. SOM-ANN has become an important tool to assess the quality of the sediments

Reviewer 2 Report

The authors provide different thermophysical properties for nanofluids with Al2O3, TiO2, and AL2O3-TiO2 nanoparticles.

The English and quality of the presentation of the manuscript need to be improved. Moreover, decide Nano or nano, Sol-gel or sol-gel. Abbreviations SOM and BP-LM algorithms should be introduced for readers.

In row 110 rho_water, in row 111 rho_w is used!

What are the size and shape of nanoparticles used in the nanofluids? These parameters strongly affect the thermophysical properties.

Reconsider the section starting in row 133.

On Fig. 3 J should be capital.

On Figs. 5, 6, 7 on the horizontal axis the temperature is in Celsius and not in Kelvin.

Please explain what is the advantage to use Artificial Neural Network for the determination of thermal conductivity over other methods. Can this method be used to produce models that are more usable in practice?

It would be good to compare the AL2O3-water and TiO2-water thermal conductivity and viscosity measured data with those reported on Fig.6a and Fig 8a.

Where and what is this nanocomposite-water mixture used for?

Author Response

Dear Editor and Reviewers,

We appreciate very much the editor and the reviewers for the constructive comments. We also thank the editor and the reviewers for the effort and time put into the review of the manuscript. Each comment has been carefully considered point by point and responded. Responses to the reviewers and changes in the revised manuscript are as follows.

Title: Prediction of Thermo-physical Properties of TiO2-Al2O3/water Nanoparticles by Using Artificial Neural Network

Reviewer 2:

  1. The English and quality of the presentation of the manuscript need to be improved. Moreover, decide Nano or nano, Sol-gel or sol-gel. Abbreviations SOM and BP-LM algorithms should be introduced for readers.

Response: The whole text is copy-edited and it is tried to correct the mistakes. All the abbreviations are defined in the revised manuscript.

  1. In row 110 rho_water, in row 111 rho_w is used!

Response: It is corrected in the revised manuscript.

  1. What are the size and shape of nanoparticles used in the nanofluids? These parameters strongly affect the thermophysical properties. Reconsider the section starting in row 133.

Response: The shape of the nanoparticles was spherical with the size of 15-30 nm.

  1. On Fig. 3 J should be capital.

Response: It is corrected in the revised manuscript.

  1. On Figs. 5, 6, 7 on the horizontal axis the temperature is in Celsius and not in Kelvin.

Response: It is corrected in the revised manuscript. The caption of Figure 6. is revised and an explanation is added to clarify that temperature is in .

  1. Please explain what is the advantage to use Artificial Neural Network for the determination of thermal conductivity over other methods. Can this method be used to produce models that are more usable in practice?

Response: ANN-based methods have this potential to give high precision estimation which can be beneficial in real practice since actual experiment is not only so expensive but also very time consuming. So, these methods can be so practical.

  1. It would be good to compare the AL2O3-water and TiO2-water thermal conductivity and viscosity measured data with those reported on Fig.6a and Fig 8a.

Response: Unfortunately we cannot add these data to the figure, but the data has been provided in a new table. The size of the nanoparticles are nearly the same. Therefore viscosity did not changed with types of nanoparticles.

Thermal conductivity

Concentration

Nanocomposite

TiO2

Al2O3

0.1

0.612

0.601

0.623

0.5

0.635

0.610

0.644

1

0.646

0.618

0.651

1.5

0.659

0.628

0.670

  1. Where and what is this nanocomposite-water mixture used for?

Response: Overall, nanofluids are used for obtaining improved heat transfer performance. In this case, this specific kind of nanocomposite water-based nanofluid, i.e. Al2O3/TiO2-water, can be used for increasing heat dissipation as the coolant fluid with improved thermal properties. This hybrid-structure is studied to achieve an enhancement in terms of thermal conductivity.

Reviewer 3 Report

Dear author, my report file is attached. you can find my opinion about your paper.

Author Response

Dear Editor and Reviewers,

We appreciate very much the editor and the reviewers for the constructive comments. We also thank the editor and the reviewers for the effort and time put into the review of the manuscript. Each comment has been carefully considered point by point and responded. Responses to the reviewers and changes in the revised manuscript are as follows.

Title: Prediction of Thermo-physical Properties of TiO2-Al2O3/water Nanoparticles by Using Artificial Neural Network

Reviewer 3:

1. In the Introduction Section, please remove the first name initial and write last name only. For example

i. In Line 40, use Vafaei et al. [22] NOT M. Vafaei et al. [22].

ii. In Line 43, use Afrand et al. [23] NOT M. Afrand et al. [23].

Response: Thanks for your attention. This comment is applied in the revised manuscript.

2. Using of TiO2-Al2O3/water Nanoparticles can be found in many applications in two-phase

flow. There are many examples of Using Artificial Neural Network in two-phase flow such as

S. Azizi, M. M. Awad, E. Ahmadloo, 2016, Prediction of Water Holdup in Vertical and Inclined

Oil-Water Two-Phase Flow Using Artificial Neural Network, International Journal of

Multiphase Flow, Volume 80, pp. 181-187, doi:10.1016/j.ijmultiphaseflow.2015.12.010.

https://www.sciencedirect.com/science/article/pii/S0301932215002797

S. Azizi, E. Ahmadloo, M. M. Awad, 2016, Prediction of Void Fraction for Gas–Liquid Flow in

Horizontal, Upward and Downward Inclined Pipes Using Artificial Neural Network,

International Journal of Multiphase Flow, Volume 87, pp. 35-44, doi:

10.1016/j.ijmultiphaseflow.2016.08.004.

https://www.sciencedirect.com/science/article/pii/S0301932216301069

Please add in Introduction Section these missing examples of Using Artificial Neural Network in

two-phase flow to the paper.

Response: The above-mentioned articles are added and cited in the revised manuscript:

35.         Azizi, S.; Awad, M.M.; Ahmadloo, E. Prediction of water holdup in vertical and inclined oil–water two-phase flow using artificial neural network. Int. J. Multiph. Flow 2016, 80, 181–187.

36.         Azizi, S.; Ahmadloo, E.; Awad, M.M. Prediction of void fraction for gas–liquid flow in horizontal, upward and downward inclined pipes using artificial neural network. Int. J. Multiph. Flow 2016, 87, 35–44.

3. In Test section, please remove extra spaces in Line 101, Figure 1 shows the schematic of

nanocomposite synthesis.

Response: Thanks for your precision. The mentioned error is corrected in the revised manuscript.

4. In Test section, please add extra spaces in Line 112, w is their mass[5].

Response: It is corrected in the revised manuscript.

5. Please add uncertainty analysis to Test section.

Response: The hot method presented a measurement uncertainty of ±2% for thermal conductivity in accordance with the manufacturer, and the deviation for each measurement was 4%. The repeatability and accuracy of the viscometer used are ±0.2% and ±1% in the full-scale range (FSR) of measurements, respectively. 

6. In Results and Discussion Section, the title of y-axis in Figure 3. The results of the specific

heat capacity of the nanocomposite should be Cp(J/gK) Not Cp(j/gK).

Response: It is corrected in the revised manuscript.

7. In Results and Discussion Section, the title of y-axis in Figure 4. The results of thermal

conductivity coefficient of nanocomposite should be k(W/mK) Not W(m/K).

Response: It is corrected in the revised manuscript.

8. In Results and Discussion Section, the title of y-axis in Figure 5. Variations of thermal conductivity coefficient of nanofluid with temperature and concentration of nanoparticles should

be k(W/mK) Not K(W/mK).

Response: It is corrected in the revised manuscript.

  1. In Results and Discussion Section, please remove extra spaces in Line 238, neurons.According.

Response: It is corrected in the revised manuscript.

10. In References Section, please write the complete list of authors names. i.e. Do not use et al.

Response: The reference format is changed according to the Journal referencing style.

Round 2

Reviewer 1 Report

The authors have addressed my concerns.

Reviewer 2 Report

After the corrections the manuscript can be accepted for publication.